# The Perception of Grassroots Coaches of Spanish Professional Clubs on the Process of Training Young Players

**DOI:** 10.3390/sports10100158

**Published:** 2022-10-18

**Authors:** Alberto Martín Barrero, Francisco Javier Giménez Fuentes-Guerra, Manuel Tomás Abad Robles

**Affiliations:** 1San Isidoro University Center, University “Pablo de Olavide”, 41092 Sevilla, Spain; 2Faculty of Education, Psychology and Sport Sciences, University of Huelva, 21071 Huelva, Spain

**Keywords:** talent, methodology, coach, social context, football initiation

## Abstract

(1) The aims of this study were to find out the perception of the coaches of male and female football players in the initiation stage of Spanish professional clubs about different factors of the training process (context, talent development, and methodology), and to identify possible differences according to age/category (U10s: 8–10 years old, and U12s: 10–12 years old) and gender (male and female). (2) For this purpose, a questionnaire of 57 questions grouped into seven dimensions was applied, selecting for this work those related to talent development, training methodology, and social context. A total of 153 coaches from top-level Spanish professional clubs participated in the study: the first and second male division and the first female division. (3) The coaches consider that genetic inheritance is more important than contextual variables in the development of sporting talent. On the other hand, they do not show a clear consensus on whether players who perform better in the initiation stage have a greater chance of reaching the elite. (4) In addition, they value decision-making and technical qualities as fundamental factors in the development of talent rather than physical preparation.

## 1. Introduction

The detection and development of talent in sports is a topic that has generated great interest and controversy among researchers for many years [1,2,3]. Traditionally, the study of these young athletes has been conducted mainly from a perspective focused on the genetic inheritance received by the individual, which determines a large part of their abilities and potential, and in which biological, anatomical, and psychological factors are highly relevant [4,5,6,7].

In recent decades, this approach has evolved towards a more global and holistic vision which considers that any ability that the individual develops after birth is a consequence of a learning process and the experiences lived, which are influenced by variables such as the type of practice, the influence of coaches, or the social and family environment [8,9,10,11,12,13]. This question has been extended to a sport as popular as football, where there is a great concern to determine which variables or factors condition the development of player talent [14,15,16]. Therefore, as Ramírez and Prieto [2] point out, it is a very dynamic process, which makes talent development a complex and multidimensional process [17,18,19] in which different factors (technical, tactical, physical, social, and psychological) intervene.

Along these lines, Sánchez-Miguel et al. [20], Lorenzo [21], Lorenzo and Calleja [22] highlight that other factors, such as contextual factors, are very important for a talented athlete to become an expert. In particular, the family becomes fundamental support in this complicated process [23,24].

The importance of methodology in the training process should also be underlined. In this sense, instructive and excessively technical teaching methods have been used in these training categories, whereas nowadays, more active and constructivist models have been used, where the players themselves, their abilities, and the context of the activities are very much taken into account [25]. Tavares [26] already stated years ago that, in the teaching of sports games, individual tactics would be the point of reference when structuring the teaching-learning process, enabling the training of players with their own decision-making capacity. Recently, different research [27,28] has shown that soccer teaching should promote diverse game contexts, sufficiently unpredictable for the opponent and sufficiently predictable for the teammates of the same team.

Meanwhile, according to Buchheit et al. [29], professional clubs are increasingly placing more emphasis on the detection and development of young talents. In this process, coaches have a great influence on the young player, becoming a key figure in the training process of the football player [30,31]. This makes it relevant to study the figure of the coach in the field of training football. In recent years, studies have been undertaken to investigate different aspects related to grassroots football from the coaches’ perception. Accordingly, numerous authors have studied the teaching methods and training of grassroots football coaches in different contexts [32,33,34]. Lledó et al. [35] focus on the profile of professional club coaches in the Community of Valencia (Spain) and conclude that competition and traditional teaching methods are still very present. On the other hand, Paixao et al. [30] studied the figure of the grassroots football coach in the province of Beja in Portugal; it is concluded that coaches use both direct teaching methods and guided learning, being sensitized to playful training methods. In turn, Pazo et al. [36] develop research on the key elements making a football player reach the elite, in which the criteria and opinions of coaches are taken into consideration.

In this research, it was deemed essential to establish the coaches as the main research subject, as their influence on the training of football players can help to approach this problem from a more holistic perspective and to obtain data to better understand the decisions and actions carried out in the development of talent. For this reason, the objectives of this study were to find out the perception of coaches of male and female football players in the initiation stage of Spanish professional clubs on different factors of the training process (context, talent development, and methodology) and to identify possible differences according to age/category (U10s: 8–10 years old, and U12s: 10–12 years old) and gender (male and female).

## 2. Materials and Methods

### 2.1. Participants

This research involved the coaches of the U10 (8–10 years old) and U12 (10–12 years old) categories of the Spanish top-level professional clubs: first and second male division and first female division (see Table 1) in the 2019–2020 season.

A total of 153 coaches participated with an average age of 30.99 ± 7.17 and distribution by gender male (*n* = 140, 91.5%) and female (*n* = 13, 8.5%) with experience in grassroots football from 4 to 7 years, with an average experience in the categories under the study of 2 years, with a specific qualification in football level II, university studies related to sport and experience as an ex-player of grassroots or amateur football. In the U10 category (8–10 years old), 59 coaches and 9 female coaches participated: 68 (44.44%), while in the U12 category (10–12 years old), 81 coaches and 4 female coaches participated: 85 (55.6%). As exclusion criteria, all those coaches who belong to the social soccer schools of professional clubs (male and female) were excluded. Inclusion criteria included i) all those coaches who belong to the youth academies of professional clubs (male and female) and ii) coaches of U10 and U12 teams.

In order to ensure that the sample was representative of the population under study, the required sample size was calculated [37]. In this process, for a population of 186 coaches, a confidence level of 99% was used, with a margin of error of 0.04 (4%) for a sample size of 153 subjects, resulting in 82.25% of the total population.

### 2.2. Instrument

The instrument used was the questionnaire designed ad hoc and validated by Martín-Barrero et al. [38], which consisted of 57 questions grouped into 7 dimensions (socio-demographic variables, talent development, training methodology, competitive context, coach qualities, coach education, and social context). The study described in this paper focused on three dimensions (24 items, 8 per dimension): (i) talent development dimension, which sought to know the variables the coach perceived to be important to detect talent and influence the talent development process; (ii) coaching methodology dimension, where the aim was to know how and which were the teaching methods, as well as their actions in training and the design and planning of the coaches’ contents; and (iii) social context dimension, which sought to obtain information on how the coaches perceived the social reality of the player in training. The questionnaire had a Likert scale with values between 1 (do not agree at all) and 10 (totally agree). For the statistical analysis, Cronbach’s alpha was calculated for each dimension studied: (i) talent development (0.768); (ii) training methodology (0.815); and (iii) social context (0.783), values that are considered acceptable.

### 2.3. Procedure

Prior to data collection, participants were informed of the purpose of the study and were assured that the information collected would be kept confidential. The study was conducted in compliance with ethical standards and the Declaration of Helsinki, and informed consent was obtained from all coaches. Furthermore, the study was approved by the Research Ethics Committee in Andalusia, Spain (Code 2138-N-20).

The research was performed in three phases: In the first phase, a list was drawn up of the teams whose training football included the U10 and U12 categories, which was verified through the heads of each club, which facilitated the compilation of the final list of coaches to be surveyed. During the second phase, the coaches were invited through face-to-face contact or by email to participate in this research. Once the coaches were informed of the objectives of the study and the voluntary nature of their participation, the questionnaire was handed out to the coaches, who were previously contacted by different means (in person, by telephone, and online platform Google Form). This was conducted during the months of October and November 2019. Once the data had been obtained, the third phase was completed, in which the data were entered into an ad hoc recording tool in an Excel sheet in Microsoft Office^®^ version 14.7.7 for Windows. Finally, the data were transferred to SPSS 25.0 (Statistical Program of Social Sciences) for analysis.

### 2.4. Statistical Analysis

A descriptive analysis was then run, where the cumulative percentages, means, and standard deviations of the items of the different dimensions analyzed were obtained. Finally, an inferential analysis was conducted between the different questions of the questionnaire and between the U10s and U12s teams, as well as between the male and female teams, in which the level of relationship and association of the different variables was reflected. The Kolmogorov-Smirnov normality test was applied, which showed the need for non-parametric tests (*p* < 0.001). For the analysis with variables from more than two groups, the Kruskal Wallis test was used with a significance level of 95% (*p* ≤ 0.05) with Bonferroni adjustment (0.05/n-1). Subsequently, the U-Mann–Whitney test was administered for pairwise comparison

## 3. Results

For a better understanding of the results, we will mainly use the cumulative percentage of each item, i.e., the percentage of the single value added to the percentage of all previous values. The tables also include the mean and standard deviation of each item (Likert scale assessment: between 1 and 10).

### 3.1. Talent Development Dimension

This dimension consisted of eight items on issues of interest for the detection and development of talent within the training process (Table 2). Firstly, it was noted that a high percentage of coaches placed great importance on the influence of genetics on player potential (60.8%) (M = 5.65 ± 1.73).

As to whether coaches considered that the players with the greatest future potential at this stage were those who performed better in competition, the results showed a slight tendency to agree with this question (51.6%; M = 5.20 ± 2.10). As for the importance coaches attributed to the physical aspect in order to identify talented players at this age, few were leaning towards agreeing with this question (28.1%; M = 4.07 ± 2.03).

Very few agreed (16.3%; M = 3.14 ± 2.35) that decision making was not a key factor in detecting talent at this stage. Furthermore, regarding taking into account the technical component as the most important component to detect talent at this stage, a high degree of agreement was found (75.8%; M = 6.75 ± 1.86). On the question of whether coaches at this stage identified goal accumulation to detect talent, there was a small percentage showing a tendency to agree with this question (15%; M = 3.30 ± 1.48).

The following question in the questionnaire was whether clubs, when selecting players in these age groups should take into account the distance players would have to travel to training and matches. Coaches slightly agreed with this consideration (54.2%; M = 5.86 ± 3.00). Furthermore, as to whether coaches considered that at this stage of training, outstanding players should leave their usual residence to be in high-performance centers or professional club residences, it was clear that very few agreed with this (10.5%; M = 2.58 ± 1.99).

### 3.2. Training Methodology Dimension

This dimension consisted of eight questions on teaching methods, design, and planning of content by coaches (Table 3).

The first question posed in the training methodology dimension raised the importance of the organization of the technical–tactical contents. The results indicated that 92.8% (M = 8.61 ± 1.45) of the coaches tended to agree with this statement. However, on the need to define a single demarcation in which the player should play in competition, few coaches agreed (19%; M = 3.11 ± 2.24). Similarly, on the question of whether players should be identified according to a playing style, 39.2% (M = 4.83 ± 2.54) it was shown some degree of agreement. Furthermore, regarding the question of whether individual coaching programs were necessary at these ages, the coaches agreed (60.1%; M = 6.09 ± 2.63).

Regarding the item on whether physical preparation contents should be worked on as an important content in the session in these categories, few coaches agreed with this statement (24.8%; M = 3.21 ± 1.72). However, in relation to whether the training contents should be oriented considering the age rather than the level of the players at this stage, the coaches showed 39.9% agreement (M = 4.99 ± 2.49). As for the item on whether the tasks being designed should be oriented more towards technical execution than problem solving, the coaches had little tendency to agree (34%; M = 4.82 ± 2.35). Finally, when asked whether the opponent and the competition should be taken into account when planning training in these categories, very few coaches agreed with this opinion (14.4%; M = 3.16 ± 2.20).

### 3.3. Social Context Dimension

In the social dimension, the aim was to obtain information on how the coaches perceived the social reality of the training player (Table 4).

On the item referring to whether coaches considered it necessary to regulate the access of family members to training sessions in these age groups, the data indicated that 73.9% of coaches agreed (M = 7.39 ± 2.66). Similarly, with regard to whether coaches felt it was necessary to regulate family members’ access during competitive matches in these age groups, respondents showed some agreement (52.3%; M = 5.75 ± 2.83).

In relation to whether coaches thought that clubs and federations provided sufficient training for players’ families, the data indicated that very few coaches agreed with this question (13.7%, M = 3.00 ± 2.13). Furthermore, to the question of whether coaches thought that parents of players at this stage were interested in receiving training to correct their misbehavior, the data reflected that 28.1% (M = 4.32 ± 2.35) of coaches had a tendency to agree.

On the other hand, when coaches were asked whether they considered that children were being introduced to football in schools and clubs at an increasingly younger age, the data indicated a high degree of agreement (83.7%; M = 7.99 ± 2.18). Concerning whether coaches considered that the higher the socio-economic status of the players’ relatives, the better their cooperation in relation to their child’s football practice, there was no clear degree of agreement. However, the data reflected that 40.5% (M = 4.90 ± 2.52) of the coaches tended to agree with this statement.

Finally, in terms of the item on whether the coaches considered that currently the attention and concern conveyed by family members to their children was excessive, the data conveyed a very clear tendency to agree with this question (90.8%; M = 8.14 ± 1.84).

### 3.4. Difference between the U10s and U12s

Below are shown the outstanding results of the difference between categories (Table 5).

The main differences between the two educational categories were found in the type of specific coaching qualification (*p* = 0.004), with a higher level of education among U12 coaches. On the other hand, significant differences were also found in the criteria related to the socio-economic level of the family (*p* = 0.021), with the U10 coaches thinking that the higher the socio-economic level of the players’ family members, the better their collaboration in relation to the practice of football with respect to their son.

### 3.5. Difference between Male and Female Clubs

Below are shown the Outstanding results of the difference between the male and female teams (Table 6).

The results showed that the male football coaches were more experienced as they accumulated more years coaching grassroots football (*p* = 0.001) and had more specific training than those coaching female clubs. Regarding the training methodology, the coaches of the female teams (*p* = 0.029) pointed out that it was fundamental for the clubs to identify their players with a certain playing style during this stage and that it was important to take into account the opponent and the competition to plan the training in these categories (*p* = 0.005). Finally, male coaches were the ones who significantly (*p* = 0.027) stated that in male clubs, parents were overly concerned about their children’s football practice.

## 4. Discussion

The aims of this study were to find out how coaches of male and female football players in the initiation stage (7-a-side football) of Spanish professional clubs perceive different variables of the training process (context, talent development, and methodology) and to identify possible differences according to age/category (U10s: 8–10 years old, and U12s: 10–12 years old) and gender (male and female).

Firstly, when the coaches were asked if they thought that genetics was the most important factor defining a player’s potential, the results showed that coaches still give too much importance to the influence of genetics in the development of talent, over contextual or environmental variables, despite the fact that numerous current studies [8,9,10,11,12,13] indicate the relevance of other intervening variables in the process (type of practice, the influence of coaches, or the social and family environment).

Along these lines, it is important to highlight that genetics will have a greater influence, especially in sports with closed and linear skills, in which the physiological and anatomical component is particularly relevant [7,39]. Nevertheless, in team sports such as football, anthropometric and physical variables are less determinant, and greater importance must be placed on cognitive aspects [21,40], such as intelligence in the game or creativity. Pazo et al. [31] identified, among the factors considered to be most important in the training process of the football player who reaches the elite level, the social context and the sporting context, which are regarded as fundamental in this process. In short, current research advocates a more holistic and contextual conception of talent development [41,42,43]. For this reason, in football, the influence of the context, together with the development of cognitive and technical–tactical skills [44,45], is a determining factor and, therefore, must be taken into account when it comes to the development of sporting talent.

As to whether the players with the greatest future potential at this stage were those who performed better in competition, the data also showed a high degree of agreement from the respondents. In this regard, it is worth highlighting the review by Fernández-Ortega et al. [46], which indicated that there are studies showing that elite athletes did not necessarily excel at an early age. On the other hand, it is worth mentioning that the effect of relative age usually conditions performance in competition until the age of 14–15 years due to differences in maturational development [47] and even in accumulated experiences [48,49,50,51]. In the study conducted by Rodríguez-Lorenzo and Martín-Acero [52] on the effect of relative age in the teams (grassroots and elite) of a professional club, it is concluded that this effect fundamentally influenced the talent selection process, especially in the younger age categories, and even the chances of debuting with the first team.

Regarding the maturational development and physical evolution of players, it was clear to coaches that they did not usually focus on the physical aspect to detect and select players at this age, as well as that decision-making ability was a fundamental aspect and that technical skills were the most important factor. Accordingly, some studies have tried to analyze the training profile of the football player from a physical perspective [53], highlighting the evolution with age of variables such as strength, speed, weight, and height. The data obtained in the research are similar to those obtained by Christensen [54], who, in his study, conducted interviews with coaches of the Danish national teams on the qualities that they considered very important to value, among which were the tactical qualities (understanding the game and decision-making) and technical skills. In addition, they also identified physical qualities as very important, although it should be noted that the coaches were not from the initiation stage. Papers such as those by Williams et al. [13] insisted on the multidisciplinary vision of talent identification and development, which must be approached from different variables, taking into account the criteria of experts and carrying out longitudinal studies.

On the other hand, when coaches were asked whether outstanding players should leave their usual residence to be in high-performance centers or professional club residences, there was a clear tendency to disagree with this statement. It is worth bearing in mind the considerations of Pruna et al. [15], who suggested that leaving home or changing one’s personal environment can increase stress levels, which can have both sporting and personal consequences. Along these lines, studies by Tiesler [55] and Bahdur and Pruna [56] also pointed out that a change in the player’s family and social environment can generate stress affecting performance, even in elite football players.

Concerning the need to have planned and organized technical–tactical content at this stage, most of the coaches agreed with this statement, which coincides with what is indicated by Lledó and Huertas [33], who considered that coaches at initiation age in professional clubs are concerned about the organization of work content. According to these same authors, this degree of involvement and preparation may be due to the greater presence of professionals with university studies in sports who take subjects related to the planning of training and the design of the teaching-learning process. In this way, Baena-Extremera et al. [57] considered it interesting to establish learning objectives and content in an organized manner, favoring autonomy and motivation.

As for the relevance of working on the contents related to physical preparation at these ages, most of the coaches disagreed with the idea of developing them during the training session. Currently, there are studies showing that these qualities can be correctly developed through game situations and small-sided games [58,59]. Nonetheless, in recent years, there has been a growing trend of the need to enhance motor coordination through situations that stimulate a greater wealth of movements [60,61] due to the lifestyle habits emerging in society, where many children do not engage in sufficient physical activity or base their “sports training” on the practice of a single sport, which may increase the risk of future injuries and reduce their chances of performance [62,63].

Regarding the need to identify players with a single demarcation and playing style, coaches disagreed on both items. In terms of demarcations, studies such as those by Sánchez-García [64] and Sánchez-García and Sánchez-Sánchez [65] reported different requirements, both from a physical point of view, both from an internal load (heart rate) and external load (kinematic variables), and from a technical–tactical point of view, depending on the demarcation and the systems of play used, and it may be favorable for the player to intervene in different demarcations during the training process before specializing. As regards the playing style, authors such as Lapresa et al. [66], González-Víllora et al. [67] pointed out that the players’ abilities must be borne in mind in relation to their maturity since, depending on their age, they have greater ease in acquiring and applying certain individual or collective concepts.

The coaches interviewed were in favor of the need to develop individual training programs for this age group. These types of training programs tend to be focused on the teaching of performance techniques. This question is related to another item in the questionnaire where they were asked about whether the design of tasks should be oriented more towards technique than problem solving, and no clear consensus was found in the answers of the respondents. These data may indicate that the coaches were not clear about how the training programs and teaching methodologies should be applied. In this regard, Oslin and Mitchell [68], González and Campos [69], and Lledó and Huertas [33] suggested that coaches at these ages usually show direct teaching methods based on the repetition of the execution of technical patterns. Furthermore, Abad et al. [32] and Paixão et al. [30] reported that coaches in grassroots football use both directive and discovery-based methods, pointing out that there may be some incongruence between what coaches think and what they do.

Concerning the training planning process, coaches showed a tendency to disagree with planning training according to the opponent and the competition, being the age and not the level of their player, as the reference for planning training. In this regard, Paixão et al. [30] found that grassroots coaches had a divided opinion on whether the objectives should be more competitive or more formative at training ages. On the other hand, and in terms of taking into account the age and not the level of the players when planning, there is a diversity of opinion in the specialized literature, as there are researchers who establish proposals based on chronological age and others based on the level of the players [70]. Nevertheless, what does seem important is to be able to temporarily adjust the contents to adapt them to the learning needs of the players, establishing planning phases and units [71].

As far as the social context is concerned, firstly, the coaches were asked for their opinion on the need to regulate the access of family members to training sessions in these age groups, and they expressed a high degree of agreement. However, when they were asked the same question but focused on the competition, i.e., the need to regulate the access of family members during competitive matches in these categories, they slightly agreed, but the percentage dropped considerably. Along these lines, Calvo-Rodríguez and García-Martínez [72] stated that parents can generate mistaken expectations about their children since many of them are unfamiliar with the sport. According to Eddy [73], parents should not see their children practicing sports if they are unable to refrain from intervening directly or indirectly in it, forgetting that their role is to support and not to pressure. Similarly, Revuelta et al. [23] and Walters [24] stated that we cannot forget that parents should be supportive and guide the integral development and enjoyment of young athletes.

The influence of parents in the training process of players is a key aspect [31]. However, are clubs and schools concerned with guiding and orienting these parents? When coaches were asked whether they considered, in general, that clubs and federations carried out sufficient training with players’ families, the vast majority strongly disagreed with this question. Mendoza [74] pointed out that, sometimes, parents, due to lack of information, harm their children’s sporting performance. In the research carried out by Lorenzo et al. [75] with approximately 100 parents in football, they tried to apply an intervention program to improve their cognitive and affective aspects and those related to guidelines and actions. The results of the study were encouraging, as they showed that, after the intervention program, parents exert less pressure on their children in the sporting environment.

Finally, another aspect studied in the social context was how lifestyle and cultural habits could influence the development of the football player. Coaches were asked whether they considered that children increasingly started playing football at an earlier age in schools or clubs, with coaches showing a high degree of agreement. Furthermore, when the coaches were asked whether they thought that children were playing less and less in the streets and more and more in clubs and schools, they showed a very high degree of agreement with this question. All this may give us food for thought about the changes in lifestyle habits currently taking place in society and how these may influence children’s activities. In fact, Baker and Horton [76] refer to the fact that the influence of a country or a certain area on a sport can facilitate the emergence of athletes who become involved in that sport. As Martín-Barrero [71] pointed out, the process of urbanization and technological development is having an impact on the lifestyle habits of the new generations, with fewer common spaces for young people to interact through motor practice.

The lack of studies addressing talent development from a holistic viewpoint which includes the analysis of the training process from the coach’s perspective, has been the main limitation. For this reason, the results should be treated cautiously, and further research is needed in this respect. As a proposal for the future, it would be interesting to extend this study to coaches of other training categories or other similar sports.

## 5. Conclusions

Coaches still consider genetic inheritance to be more important than contextual variables in the development of sporting talent. On the other hand, they do not show a clear consensus on whether players who perform better in the initiation stage have a better chance of reaching the elite. Furthermore, they value decision-making and technical qualities as fundamental factors in the development of talent and not so much physical preparation.

On the other hand, the coaches consider the planning and organization of the technical–tactical content to be very important. They also state that during the initiation stage of football, the player should not be identified with a single demarcation on the pitch nor with a playing style, which is not the case in female teams. Moreover, in relation to the planning of training, the coaches state that it should be planned according to the age of the players they train and not so much according to their level.

As regards the social context, it can be concluded that coaches consider it very important to regulate the access of family members to training sessions, although not so much to competitions. As for the training, parents can be offered for the education of their children, and coaches indicate that clubs and federations do not show any concern for training them. Furthermore, they warn that parents are increasingly enrolling their children in football schools at an earlier age. Finally, the coaches of the male teams perceive that, currently, the attention and concern that family members convey to their children in relation to the practice of football is excessive, while this is not the case in the female teams.

As practical proposals, the need to include content related to the contextual variables determining the development of talent, as well as practical interventions where future coaches can experience and learn the new methodological trends in sports education, in the training program of the football federations’ coaching schools and in Sports Science studies, is highlighted. Finally, it is considered necessary for clubs to develop methodological proposals focused on girls in order to improve their training process and increase the value of female football.

## Figures and Tables

**Table 1 sports-10-00158-t001:** Clubs and coaches participating in the research.

1st Male’s Division		2nd Male’s Division		1st Female’s Division	
Club	N	Club	N	Club	N
Ath. Bilbao	2	Huesca	3	Atlético de Madrid	6
Atlético de Madrid	4	Cádiz	2	F.C. Barcelona	2
F.C. Barcelona	5	Rayo Vallecano	4	Real Betis	1
Real Betis	8	Real Zaragoza	2	Sevilla F.C.	1
Sevilla F.C.	8	Almería	2	Valencia C.F.	1
Valencia C.F.	4	Girona	4	Levante	4
Villareal	5	Elche	3	Espanyol	1
Osasuna	3	Fuenlabrada	3	Tacón	1
Eibar	1	Las Palmas	2	Sporting Huelva	2
Valladolid	2	Alcorcón	4	Madrid CF	6
Levante	4	Tenerife	2		
Real Madrid	4	Sporting de Gijón	3		
Leganés	4	CD Lugo	4		
Mallorca	4	Albacete	1		
Espanyol	4	Ponferradina	4		
Celta de Vigo	2	Deportivo de la Coruña	4		
		Numancia	4		
		Extremadura	3		
		Racing de Santander	2		
		Málaga	4		
		Real Oviedo	4		

**Table 2 sports-10-00158-t002:** Outstanding results of the talent development dimension.

Variable	Cumulative %	M	SD
I think that the genetic basis is the most determining factor in defining a player’s potential.	60.8	5.65	±1.73
I believe that the players with the greatest future potential at this stage are those who perform best in competition.	51.6	5.20	±2.10
I tend to look at the physical aspect when spotting talented players at this age.	28.1	4.07	±2.03
I consider that decision making is not a key factor in identifying talent during this stage.	16.3	3.14	±2.35
For me, technical qualities are the most important component when it comes to spotting a player’s talent at this stage.	75.8	6.75	±1.86
I usually take into account, in my criteria to detect players at this stage, the number of goals they have accumulated in the different competitions.	15.0	3.30	±1.48
I consider it essential that clubs take into account the number of kilometers they need to travel in order to train and play when selecting players in the U10 and U12 categories.	54.2	5.86	±3.00
I think that “at this stage of training,” outstanding players should leave their usual residence to stay in high-performance centers or in residences of professional clubs.	10.5	2.58	±1.99

**Table 3 sports-10-00158-t003:** Outstanding results of the training methodology dimension.

Variable	Cumulative %	M	DT
I consider it important to have the technical–tactical contents organized at this stage.	92.8	8.61	±1.45
I think it is important to establish a single demarcation in which to play in competition at this stage.	19.0	3.11	±2.24
I think it is essential that clubs identify their players with a particular playing style during this stage.	39.2	4.83	±2.54
It is necessary to have individual training programs with players at this stage.	60.1	6.09	±2.63
In this category, physical preparation content should be worked on as important content in the training session.	24.8	3.21	±1.72
I believe that the content of training should be oriented according to the age rather than the level of the players at this stage.	39.9	4.99	±2.49
I believe that during this stage, the tasks designed should be oriented more towards technical execution than problem solving.	34.0	4.82	±2.35
I think it is important to take into account the opponent and the competition when planning training in these categories.	14.4	3.16	±2.20

**Table 4 sports-10-00158-t004:** Outstanding results of the social context dimension.

Variable	Cumulative %	M	SD
I think it is necessary to regulate the access of family members to training sessions at this age.	73.9	7.39	±2.66
I consider it necessary to regulate the access of family members during competitive matches in these categories.	52.3	5.75	±2.83
I believe that, in general, clubs and federations provide sufficient training for players’ families.	13.7	3.00	±2.13
I think that parents of players at this stage are interested in receiving training to correct their misbehavior.	28.1	4.32	±2.35
I believe that children are being introduced to football at an increasingly younger age in schools and clubs.	83.7	7.99	±2.18
I think football is being played less and less in the streets and more and more in schools and clubs.	94.8	8.90	±1.71
I believe that the higher the socio-economic level of the players’ relatives, the better their cooperation in relation to football practice with regard to their son/daughter.	40.5	4.90	±2.52
I believe that currently, the attention and concern that family members convey to their children in relation to football practice is excessive.	90.8	8.14	±1.84

**Table 5 sports-10-00158-t005:** Outstanding results of the difference between categories (U10/U12).

According to the Category in Which They Train	Assint. Significance
Years of training in the U12 category	0.001
Years of training in the U10 category	0.000
Specific coaching qualification	0.004
I believe that the higher the socio-economic level of the players’ relatives, the better their cooperation in relation to football regarding their son/daughter	0.021

**Table 6 sports-10-00158-t006:** Outstanding results of the difference between the male and female teams.

According to the Gender of the Team (Male or Female)	Assint. Significance
Age	0.005
Years coaching grassroots football	0.001
Specific coaching qualification	0.040
Experience as a player	0.001
I think it is essential that clubs identify their players with a particular playing style during this stage.	0.029
I think it is important to take into account the opponent and the competition when planning training in these categories.	0.005
I believe that currently, the attention and concern that family members convey to their children in relation to football practice is excessive.	0.027

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
