# Peer review of "The Perception of Grassroots Coaches of Spanish Professional Clubs on the Process of Training Young Players"

_sports, 2022, doi:10.3390/sports10100158_

Round 1

Reviewer 1 Report

It articulately describes the procedure and the analysis of the results, which is very good. Although it is not a descriptive study, the authors delve deeper into an inferential statistical analysis.

The inclusion and exclusion criteria are missing in the description of the simple.

Author Response

Exclusion and inclusion criteria have been included.

Reviewer 2 Report

First, I would like to congratulate the authors for their work and for the relevance of the problem under study.

I have a few small recommendations:

- Line 30 - instead of [4, 5, 6, 7] should be [4-7]. Fix all.

- Remove spaces in references

- What practical implications can the results have (despite the limitations) in terms of the vision that is taken on the detection and development of talents.

Author Response

Modified Line 30 - instead of [4, 5, 6, 7] should be [4-7]. Fix all.

Modified remove spaces in references

Practical implications included.

Reviewer 3 Report

First of all, I would like to thank the authors for the presented results of their investigation on football coaches. Also, I would like to thank the editor for the opportunity to review this manuscript.

In a study of grassroots coaches, the authors of a manuscript titled "The perception of grassroots coaches of Spanish professional clubs on the process of training young players" compare and contrast coaches' perspectives on training for male and female football players at the youth and developmental levels at professional clubs in Spain. In my opinion, the authors present an interesting topic that falls within the aims and scope of the Sports journal.

The Introduction section should provide more background regarding coaches of young football players, specificity and differences between the 8-10 and 10-12 age categories, as well as specificity and differences between boys and girls. I believe that improvements in the Introduction part could lead to more clarity in the Methods. Apart from that, the Methods should provide some more explanations about the sample.   

In particular:

Abstract

Line 10: I would suggest some alternative term instead variables, e.g., factors.

Introduction

Line 43: I would suggest avoiding many studies and citing three references.

Lines 45, 46: References are not in line with the Instructions for Authors.

Line 48: Which training process?

Lines 49-52: This sentence is too wide.

Lines 64-68: These sentences are redundant. Instead of what was their focus on, it should be introduced what they found.

Line 75: I would suggest some alternative term instead variables, e.g., factors.

Materials and Methods

Participants

Lines 81-83: What was the rationale for choosing the presented clubs? Are "the first and second male's division, and first female's division" related to seniors' teams? If so, the season when the investigation was conducted should be presented because these data are not valid for the season 2022/2023.

Table 1. Ath. Bilbao is listed in the 1st Female’s division without the number.

Lines 89-99: The data should not be duplicated in the text and table.

Instrument

Line 109: Space reasons should not be justification in the scientific paper.

Statistical analysis

Line 146: I suggest using p < .001

Line 147: I should be p ≤ .05.

Results

Lines 151-154: This is redundant.

Generally, the results should not be duplicated in the text and table.

Discussion

I would suggest using the past tense when referring to the previous research findings.

Lines 292-294: How is this statement connected with the results of this research?

Line 335: The number should be in the brackets.

Line 401: I would suggest using general form, meaning to avoid using we.

Author Response

Revisor 3

In particular:

Abstract

Line 10: I would suggest some alternative term instead variables, e.g., factors.

Introduction

Line 43: I would suggest avoiding many studies and citing three references.

Lines 45, 46: References are not in line with the Instructions for Authors.

Line 48: Which training process?

Lines 64-68: These sentences are redundant. Instead of what was their focus on, it should be introduced what they found.

Line 75: I would suggest some alternative term instead variables, e.g., factors.

Materials and Methods

Participants

Lines 81-83: What was the rationale for choosing the presented clubs? Are "the first and second male's division, and first female's division" related to seniors' teams? If so, the season when the investigation was conducted should be presented because these data are not valid for the season 2022/2023.

Table 1. Ath. Bilbao is listed in the 1st Female’s division without the number.

Lines 89-99: The data should not be duplicated in the text and table.

Instrument

Line 109: Space reasons should not be justification in the scientific paper.

Statistical analysis

Line 146: I suggest using p < .001

Line 147: I should be p ≤ .05.

Results

Lines 151-154: This is redundant.

Generally, the results should not be duplicated in the text and table.

Discussion

I would suggest using the past tense when referring to the previous research findings.

Lines 292-294: How is this statement connected with the results of this research?

Line 335: The number should be in the brackets.

Line 401: I would suggest using general form, meaning to avoid using we.

Line 10: included “factors”.

Citing three references.

Mistake corrected.

Line 48: the term "this" is modified by the term “the”

Modified and include the main conclusion what they found.

Included “factors”.

Líneas 81-83: incluided season.

Mistake corrected.

In order not to duplicate the data, table 2 has been deleted and missing information has been added to the text.

Mistake corrected.

Modified

Modified

Although we agree with the reviewer’s suggestion, we believe that the inclusion of the tables helps to better understand the information contained in the text and the reading of it.

Modified

Modified and justified.

Modified.

Modified.

Reviewer 4 Report

Dear Authors,

I have a very pleasant impression after reading the article. Indeed, the role of the coach in the development of sports talents is decisive, accordingly, understanding the thoughts, attitudes and decisions of the coach, taking into account the social and cultural context can provide a lot of interesting information about both the strengths of the approaches they choose, and identify weaknesses that can be worked on in the future. Coaches who work with age groups of 8-12 years are really focused on finding and developing sports talent, their contribution is very important in the development and formation of an elite player. In my opinion, the work is relevant, the research is well planned and described, and the discussion is sufficiently thorough. I would like to mention a few recommendations that will help some of the issues in the article become more clear.

1. It is worth adding some additional data related to the organization of the study, namely when exactly the survey was conducted, how exactly it was organized (for example, if online, then with which platform), and how the respondents were invited to the study, whether everyone agreed to take participation in it, whether there is a certain proportion of persons who refused it, etc.

2. It is worth adding more modern scientific literature to the introduction or discussion, in my opinion, there should be more articles from the last 5 years in the list of references. This will only strengthen the novelty of your research, and possibly add new meanings to it.

3. Information related to Cronbach's alpha of different parts of the questionnaire (lines 138-140) is more relevant to the Instrument subsection (line 105). Perhaps it is worth in this subsection that noting that Cronbach's alpha of the questionnaire within the scope of this study is exactly the one that corresponds to the permissible value [REF]

Kindest regards,

Author Response

Revisor 4

1. It is worth adding some additional data related to the organization of the study, namely when exactly the survey was conducted, how exactly it was organized (for example, if online, then with which platform), and how the respondents were invited to the study, whether everyone agreed to take participation in it, whether there is a certain proportion of persons who refused it, etc.

2. It is worth adding more modern scientific literature to the introduction or discussion, in my opinion, there should be more articles from the last 5 years in the list of references. This will only strengthen the novelty of your research, and possibly add new meanings to it.

3. Information related to Cronbach's alpha of different parts of the questionnaire (lines 138-140) is more relevant to the Instrument subsection (line 105). Perhaps it is worth in this subsection that noting that Cronbach's alpha of the questionnaire within the scope of this study is exactly the one that corresponds to the permissible value [REF]

Line 151: Added namely when exactly the survey was conducted.

Line 151: Added how exactly it was organized (for example, if online, then with which platform)

Line 147: Added how the respondents were invited to the study.

New references added for the last 5 years introduction and discussion

Information related to Cronbach’s alpha from different parts of the questionnaire has been changed to paragraph as suggested by the reviewer

Round 2

Reviewer 3 Report

I would like to thank the authors for their effort in improving the manuscript.